# The Role of Baptism in Christian Identity Formation

Michael D. O'Neil

Morling College, Perth (Vose) Campus, Bentley, WA 6102, Australia; michaelo@morling.edu.au

**Abstract:** The construction of one's identity in late modernity is sometimes viewed as a project of the autonomous self in which one's identity may shift or change over the course of one's existence and development. For the Christian, however, one's identity is both a divine gift, and a task of ecclesial formation, and for both the gift and the task, Christian baptism is fundamental. Baptism represents the death of the self and its rebirth in Christ, a decisive breach with the life that has gone before. Baptism establishes a new identity, a new affiliation, a new mode of living, and a new life orientation, direction, and purpose. This paper explores the role of baptism in the formation of Christian identity, finding that Christian identity is both extrinsic to the self and yet also an identity into which we are called and into which we may continually grow. The essay proceeds in three sections. It begins with a survey of recent philosophical reflection on the concept of identity, continues by reflecting on the nature of Christian baptism in dialogue with this reflection, and concludes by considering in practical terms how baptism functions in the process of conversion–initiation toward the formation of mature Christian identity.

**Keywords:** identity; Christian identity; Christian formation; baptism; conversion–initiation; Social Identity Theory

## 1. Introduction

Pressed by the Roman proconsul to repent, to deny Christ, and to swear instead in Caesar's name, and so pretend "not to know who and what I am", Polycarp, bishop of ancient Smyrna, answered, "hear me declare with boldness, I am a Christian". His bold confession expresses the weight of a life lived in worship and service of Jesus Christ: "Eighty and six years have I served Him, and He never did me any injury: how then can I blaspheme my King and my Saviour?"[1]

Not every believer in Jesus Christ either then or now is called upon to assert and defend their Christian identity in martyrdom, as Polycarp did. Nevertheless, although the martyrs' fate might not be ours, their faith and conviction must be (Sittser 2007, p. 48). The martyrs chose Jesus Christ—above self, family, and security, above Rome, and above life itself. Their martyrdom was a consequence of their commitment and witness rather than a quest, a death-wish, a heroic spirituality, or a theology of self-sacrifice. For Gerald Sittser, martyrdom is foundational to Christian spirituality because it highlights what is distinctive and essential in Christian faith: the sole lordship of Jesus Christ, and the utter truthfulness of his gospel (Sittser 2007, pp. 28, 47). The point is not to die a martyr's death—unless we find we are called to that—but to live a martyr's life as a witness to Christ (Sittser 2007, p. 31). Such a challenge, however, requires the inculcation and nurture of the kind of Christian identity we see displayed in Polycarp's witness, whose confession indicated a congruence between his internal self-concept and its public expression, his Christian identity sufficiently robust to hold him steady in the face of public denunciation, and the likelihood of excruciating death.

It is unlikely that anyone could write a definitive text on Christian identity formation applicable in all cases.[2] The reasons are simple. First, such formation will necessarily vary to some degree from Christian to Christian in accordance with their context and experience;

and second, even if certain common patterns of experience in formation are discernible, the work of the Holy Spirit in each person's life resists the imposition of a one-size-fits-all approach to such formation. Thus, my aim in this essay is modest; I explore the relation between baptism and personal identity, and argue that this practice has a central role to play in the formation of a distinctively Christian identity. I approach my task in three stages. Given the prominence and somewhat fractious discussion of the concept of identity in popular discourse, I begin by surveying recent reflection on the nature of identity to establish the context and some parameters for thought about Christian identity. Next, I suggest some of the ways in which Christian baptism both addresses and challenges the issues emerging in the initial survey. Finally, I consider practical implications of the discussion for Christian formation in a congregational context.

## 2. What Is Identity?

The notion of identity is complex. In their review of perspectives garnered from recent research, Crawford and Rossiter suggest that one's sense of identity is shaped through a dynamic interplay of no fewer than five centres of influence, including popular culture, distinctive ethnic and/or religious heritage, national identification, personal needs, interests, and ambitions, and family and adolescent friendship groups.[3] They list twenty-one 'components' or 'dimensions' of personal identity, including such things as one's name, gender, sexual identity, core values and moral code, age, dress, work, and so on (Crawford and Rossiter 2006, pp. 92–94). The list is not intended to be exhaustive. They note multiple and sometimes incompatible psychological theories of identity and identity formation, all of which propose that the construct is fundamental for understanding a person, their motivations, and behaviour (Crawford and Rossiter 2006, pp. 106–14). Crawford and Rossiter define personal identity as a "working hypothesis of the self . . . a process in which individuals draw on both internal and cultural resources for their self-understanding and self-expression" (Crawford and Rossiter 2006, pp. 116, 124). The idea that one's identity is a 'working hypothesis of the self' does not mean that one's identity is entirely malleable or at the whim of the individual subject. They contend, rather, that one's identity is relatively permanent and stable, but also open to gradual modification across the life cycle, resulting from new experiences or other inputs. The integration of ideas, beliefs, values, and images is used as a resource to make sense of one's life and experiences, and to shape one's self-expression. The authors also insist that identity formation must be concerned not merely with *process* but must include both process and content (Crawford and Rossiter 2006, pp. 113–14). The distinction is important, they argue, because identity may be more, or less, healthy. Their argument is with educationalists who suggest that "Education should therefore not aim at identity-development or identity-formation, but at rational autonomy, independence and responsibility, the capacity to make informed choices or at personhood".[4] In this view, it is not the role of others to hand on a particular cultural identity, especially to children, teenagers, and young adults. If cultural or religious traditions are to be considered at all, it is not to instil the cultural or religious identity in the learner, but to provide a range of possible materials that a person may or may not adopt "in their own idiosyncratic personal quest for meaning and identity". Crawford and Rossiter are concerned not merely with the epistemological presuppositions underlying this approach, but also with the idea that identity is somehow morally neutral (Crawford and Rossiter 2006, pp. 122–24).

While the general principle of being respectful of all identities is an important one, this democratic ideal has limits protected by law—we should not be equally tolerant of identities that clearly compromise the rights and freedoms of other people. This principle is also important when examining relationships between identity and violence (Crawford and Rossiter 2006, p. 119).

Identity formation has moral content and should in principle be open to moral evaluation.

Finally, a mature identity maintains a relatively harmonious balance between internal/personal and external/cultural identity resources, while being based primarily on

internal resources such as beliefs, values, and commitments. A person may indeed find support and reference points for their identity in their external relationships and cultural resources but not be so dependent on external affirmation that their own autonomy is compromised. A person with a strong sense of personal identity has not adopted their identity in an unreflective manner but has chosen it for themselves and increasingly identified with it in their own self-understanding and expression (Crawford and Rossiter 2006, pp. 121, 125; see also pp. 93–94).

Another psychological approach, Social Identity Theory (SIT), proposes that identity is shaped by an individual's participation in particular social groups. Social identity is understood as "that part of an individual's self-concept which derives from his knowledge of his membership of a social group (or groups) together with the value and emotional significance attached to that membership".[5] This definition notes the *personal* rather than corporate nature of this identity, and includes three discrete aspects: the cognitive awareness that one belongs to the group, an evaluative component that such belonging carries positive or negative value connotations, and a corresponding emotional component in which the individual may experience a range of emotional responses with respect to their own group, and towards others who stand in certain relations to it (Esler 2014, p. 17). At the heart of SIT lies an explanation of the tendency for the members of one group to compare their group to other groups, in order to achieve positive distinctiveness in relation to them (Esler 2014, p. 28).

Initiated originally by Henri Tajfel in 1978, SIT was concerned with understanding intergroup relations and conflict. Tajfel's experience of surviving the Second World War because his German captors considered him a French rather than a Polish Jew—that is, they 'categorised' him as a member of a 'group'—provided grounds for the rejection of a reductionist tendency in social psychology which considered all group interactions as 'essentially and entirely a psychology of individuals' (Esler 2014, pp. 15–16; see also Hogg 2018).

> The extraordinary discovery made by Tajfel and his colleagues, and repeated often in subsequent studies, was that the mere fact of 'social categorization', of being included in a group, led to intergroup behaviour that discriminated (via the distribution of rewards and penalties) against the outgroup and favoured the ingroup. (Esler 2014, p. 14; See also Tajfel et al. 1971, p. 172)

Some who followed Tajfel generated subsidiary theories, deepening and extending his original insights. Whereas SIT primarily explored intergroup relations and the propensity of groups to differentiate on the basis of positive comparative social identity, Tajfel's student and later collaborator John Turner, developed what he called Self-Categorisation Theory: an explanation of how a collection of individuals come to perceive and define themselves and to act as a single unit, feeling, thinking, and self-aware as a collective entity (Esler 2014, p. 24). Two processes facilitate this transition. First, individuals *self-categorise*; they become aware that they have greater affinity with some people or social category than others, thereby internalising preformed culturally available information ("I am a girl", or, "I am Catholic"). When two or more persons in a given situation share a common self-categorisation, they tend to form a group. Second, the members of the group *depersonalise*, which Turner and his associates define as 'self-stereotyping'; their self-concept is conditioned by their group membership. They tend to view themselves not merely as individuals but as exemplars of the particular social category to which they belong. Depersonalisation does not connote the loss of personal identity, nor the submergence of the self in the group. Rather, depersonalization allows the individual to *gain* identity, to live and act in accordance with broader social and cultural similarities and differences built up in human cultures over time. It indicates a shift from a personal to a social level of identity—a concept of central importance. Membership of the group must become *salient* in the person's self-concept—"cognitively prepotent in self-perception to act as the immediate influence on perception and behaviour" (Esler 2014, pp. 24–25). An important aspect of salience is the psychological ingroup ties, the "emotional merging of self with others", by

which group members gain a sense of attachment to and belonging with others in the group (Cameron 2004, p. 242).

Social Identity Complexity Theory, another development pioneered by Sonia Roccas and Marilynn B. Brewer, attempts to overcome a problem identified in SIT; persons belong to multiple groups simultaneously, thus generating multiple dimensions of identity. For example, a person may be white, English, female, a lawyer, a Cambridge University graduate, a member of the local tennis club, and a mother (Roccas and Brewer 2002, p. 88). Where significant overlap between ingroups is perceived (for example, "to be Thai is to be Buddhist"), identity complexity is reduced. Where, however, ingroups are non-converging ("I am Thai, and a Christian"), identity and social representation become more complex (Roccas and Brewer 2002, pp. 89–90). Roccas and Brewer provide four models to map social representation in complex situations. Persons may identify solely with (i) the *intersection* of their primary ingroups ("I am a Thai-Christian") regarding non-Christian Thais or Christians in general as an outgroup. Or one identity might (ii) *dominate* or even subsume the other, or a person may (iii) *compartmentalise* their social representations according to their context. Finally, they may (iv) *merge* both identities under the umbrella of an overarching or more encompassing identity ascription ("I am a Thai-Christian living and working in the United States: I am an *American* Thai-Christian").[6] Identity complexity is deepened by a range of additional social, personal, and situation factors such as overall social complexity, the motivational needs or personal values someone holds, the distinctiveness of the groups one belongs to, or the threats perhaps faced by the groups (Roccas and Brewer 2002, pp. 95–99).

SIT researchers note that a distinctive feature of group conformity concerns the role of group norms, those "regularities in attitudes and behavior that characterize a social group and differentiate it from other social groups". Group norms distinguish this group from another, provide order, predictability, and shared standards for appropriate behaviour, thus helping to enhance and maintain group identity.[7] Groups assist new members integrate when leaders especially, as well as other group members, embody the prototypical characteristics of the group (Esler 2014, p. 34). They also assist new members by offering social support and need satisfaction (Esler 2014, p. 26). Such norms may include patterns of beliefs. New members to a group are helped most when a group's norms, values, and beliefs are held confidently because they are considered true and as such are central to the group's life and decision making (Esler 2014, p. 35).

When social identity is salient, the members will align themselves with group norms, they will have been provided with a common perspective on reality, they will be motivated to coordinate their behaviour around group norms, and they will work collaboratively to further their collective self-interest (Esler 2014, p. 34).

Approaching the topic from a philosophical rather than psychological perspective, Francis Fukuyama has argued that the root of the quest for identity lies in Plato's 'third part of the soul', in *thymos*, which he describes as the cry for recognition and dignity. By grounding this quest in the soul, Fukuyama makes it a permanent aspect of human nature, an innate desire. For Plato, *thymos* was not evenly distributed but accrued to those who risked their lives for the public good, that is, the city guardians and warriors (Fukuyama 2018, pp. 18, 20–21). In modernity, however, Romanticism valorised the solitary individual in the depths of whose being lay their authentic self, their true identity, unacknowledged and even at odds with the surrounding society that demands conformity to its standards, rules, and authorities. In the twentieth century, this concept morphed from a merely personal quest into a political and social agenda, so that by 1992, U.S. Supreme Court Justice Anthony Kennedy could argue that liberty entails "the right to define one's own concept of existence, of meaning, of the universe, and of the mystery of human life" (Fukuyama 2018, p. 55). Since then, the quest for one's own identity has extended to become a personal, political, and moral imperative; an individual's self-actualisation has a higher value than the requirements of the broader society (Fukuyama 2018, p. 93). In the therapeutic culture that arose and flowered in the twentieth century, one's happiness was

understood as dependent upon one's self-esteem, and one's self-esteem was a by-product of public recognition (Fukuyama 2018, p. 100).

Another philosopher, Kwame Anthony Appiah, speaks of identity as having both an objective and a subjective dimension. Objectively, identities are labels assigned to us by others based on generally observable characteristics interpreted through a set of culturally prescribed categories (Appiah 2018, p. 5). The most basic of these labels, for example, is whether one is recognized ('labelled') at birth as either a boy or a girl. But many other labels might be applied over the course of one's life such as *Italian* (nationality), *working-class* (class), *Goth* (youth sub-culture), *Muslim* (religion), *black* (colour), *Asian* (race), *lesbian* (sexuality), *Republican* (political affiliation), *indigenous* (heritage), *champion* (achievement), or *ex-con* (social status). Such labels provide a sense of where and how one fits—or does not fit—within the social world, as well as indications of how one is expected to think and behave. Identity ascriptions may also be used by others as an indication of how they may or should treat the one who bears the identity (Appiah 2018, pp. 8–12). An identity, however, cannot arbitrarily be imposed upon another, but must be accepted, identified with, and acted upon: the subjective dimension of identity. Further, says Appiah, identity is always plural, and the shape of one's identity can also be contoured by one's other identities; the idea of intersectionality applies to every person (Appiah 2018, pp. 17–20). At the base of the labels and stereotypes that constitute identity ascriptions is the idea that these labels describe 'things of the same kind', some underlying reality or 'essence' that gives a person their true nature and so constitutes their identity (Appiah 2018, pp. 21, 25–29). This, for Appiah, is *The Lie that Binds* and the reason he has written his book, arguing that the way in which contemporary culture thinks of identity must be rethought. Whether the issue is religion, nationality, race, class, or culture, Appiah rejects a purist approach that seeks an illusory essence or singular interpretation for an identity ascription: "people have supposed that an identity that survives through time and across space must be underwritten by some larger, shared commonality; an essence that all the instances share. But that is simply a mistake" (Appiah 2018, p. 199; cf. pp. xvi, 29, 113–22). Identities are grounded not in essence but existence:

> The existentialists were right: existence precedes essence; we are before we are anything in particular. But the fact that identities come without essences does not mean they come without entanglements. And the fact that they need interpreting and negotiating does not mean that each of us can do with them whatever we will. For these labels belong to communities; they are a social possession. (Appiah 2018, p. 217)

Rather than being grounded in an illusory essence, identities are communal, historically conditioned, and can be understood and maintained by means of narrative rather than essence (Appiah 2018, pp. 65, 199). Essentialism, he insists, is a conceptual error underwriting a moral error which leaves us divided and alienated from other human beings, isolated and confined (Appiah 2018, p. 218). Appiah argues that there is only one identity which ultimately should bind all people: our common humanity, though held with a cosmopolitan ethos that is willing and able to accept that there are in fact myriad ways in which human persons can live out this common, universal identity.[8]

A surprising voice in this discussion is that of Stan Grant, a prominent Australian journalist, writer, and a Fellow of the Academy of the Social Sciences in Australia. In his 2019 book *Australia Day*, Grant speaks of the "crazy mix of DNA that makes me who I am", writing that he has "come to be suspicious of that word, identity" (Grant 2019a, pp. 53–54).

> I am an Australian—yet my history tells me that my sense of citizenship and belonging is fragile and fraught. I belong to a nation; I belong to family and a people and yet I am an individual free to determine for myself who and what I wish to be. . . . Yet the freedom to choose was taken from me when Australia had already settled on what I was: black, a half-caste, an outcast; I was not born

into Australia. My identity was already determined and I have spent a lifetime
working my way free. (Grant 2019a, p. 54)

Grant's comment illustrates those features identified by Fukuyama and Appiah. Finding himself ascribed an objective identity by others—one which constrained, isolated, and to some degree at least, determined his existence—Grant fought for recognition, for a self-chosen, liberating identity. Born to a white mother and an indigenous father, he nonetheless takes an acerbic view toward the idea of identity, referring to it as a 'poisonous new faith' (Grant 2019b, p. 91), a 'prison-house of our imaginations' (Grant 2019b, p. 58), something that binds rather than liberates, for the expectations of identity are soul-eroding, stultifying, and 'annihilating' (Grant 2019b, pp. 43, 57). One cannot be free, one cannot love, if one capitulates to the contemporary demands of 'identity'. "How easily [identity] morphs into tyranny" (Grant 2019b, p. 26)!

Enough of the toxic, political imperative of identity—the identity warriors can have that; I am done with it. Identity, even with the best of intentions, falls too easily into the hands of petty tyrants—those identity police who monitor our words and actions, trolling social media to keep people in their lanes, telling us who qualifies. . . . Identity carves us up and sets us against each other. (Grant 2019b, pp. 83–84)

It is evident that Grant is speaking about the demands of contemporary identity politics, and he does so with reference to both the left and the right of the political spectrum. He has in view, similar to Appiah, the acceptance or imposition of a 'singular' identity. His reaction derives from his own experience as an indigenous Australian who is constantly asked to tick a box affirming that he is (or is not) an 'Aboriginal or Torres Strait Islander'. To tick the box, however, is to deny his grandmother. If his son ticks the box he must erase, as it were, his mother. "It confirms what I have come to believe is true: identity—exclusive identity—has no space for love" (Grant 2019b, p. 28).

One of the more important voices in contemporary reflection on identity is that of Charles Taylor, whose life's work has been that of excavation, an endeavour to uncover the sources and character of the contemporary self. In part one of *Sources of the Self*, "Identity and the Good", Taylor argues for a 'transcendental' grounding of human identity, noting that the conditions of human personhood are that persons are situated in a moral context, in community, and in time. With respect to the first of these conditions, Taylor insists that identity is axiological. The self is set and exists in a pre-existing and pre-ordered moral 'space' in which questions are put to the agent about what is good, valued, worthy, etc. One's identity is shaped by one's answers to these questions.

To know who I am is a species of knowing where I stand. My identity is defined by the commitments and identifications which provide the frame or horizon within which I can try to determine from case to case what is good, or valuable, or what ought to be done, or what I endorse or oppose. In other words, it is the horizon within which I am capable of taking a stand. (Taylor 1989, p. 27; cf. 77–78)

Likewise, one's identity is shaped by belonging to a linguistic community in which language shapes our apprehension of ourselves in the context of others and the world. To be a 'self' is to be embedded in a (transcendent) 'web of interlocution'—dialogue and conversation—that defines us as persons. This remains true even if one experiences a decisive shift in their sense of identity: "Even the most independent identity cannot step outside of the human condition: they still elaborate their new identity and perspective by means of language and in webs of interlocution" (Taylor 1989, p. 37). One's identity is shaped, thirdly, by the teleological nature of the moral space within which human life is situated. Taylor explains that one's life in its totality and wholeness is oriented or pivoted toward the Good as they perceive and affirm it. This is a matter of allegiance to the Good, of *more* or *less* conformity to or apprehension of this Good. Just what this Good will be is connected "in a complex way with our being *moved* by it" (Taylor 1989, p. 73, original

emphasis). Further, one's life is also set in direction of the Good, a firm, settled direction and commitment in which we are *becoming* or *moving* or *going*. This twofold orientation toward the Good helps us make sense of our life by constituting our identity in terms of a narrative or quest that unifies our sense of self through time (Taylor 1989, pp. 41–48).

> I have been arguing that in order to make minimal sense of our lives, in order to have an identity, we need an orientation to the good, which means some sense of qualitative discrimination, of the incomparably higher. Now we see that this sense of the good has to be woven into my understanding of my life as an unfolding story. But this is to state another basic condition of making sense of ourselves, that we grasp our lives in a *narrative*. (Taylor 1989, p. 47; original emphasis)

Taylor's argument in this section of his book is based on a form of moral phenomenology, an attempt to give a 'Best Account' that makes sense of human life and the choices we make (Taylor 1989, pp. 58–59). Moral phenomenology refers to the lived experience of the human agent in which certain moral intuitions are inescapable, as is the language used to describe them. Taylor contends that this lived experience points inexorably to the three 'transcendental conditions' he has posited as necessary for a sense of personal identity that makes sense of one's life and experience. One must be oriented toward the Good, however it is perceived, if one is to be a self 'with an identity' (Taylor 1989, p. 68).

Turning to contemporary conceptions of the Western self, Taylor finds at the heart of modern secularity an 'exclusive humanism', a view of personal and social life—a 'social imaginary'—in which the idea of God is not necessary as an explanatory tool for understanding life, the world, and the cosmos, and in which, therefore, the highest Good has to do with the nature and pursuit of human flourishing; there is no higher goal nor any allegiance to anything else beyond this flourishing (Taylor 2007, pp. 15–19). A 'social imaginary' is the "way ordinary people 'imagine' their social surroundings, and this is often not expressed in theoretical terms, it is carried in images, stories, legends, etc". It is that "common understanding which makes possible common practices, and a widely shared sense of legitimacy" (Taylor 2007, pp. 171–72). Further, according to Taylor, we live in 'an age of authenticity' in which "each one of us has his/her own way of realizing our humanity, and that it is important to find and live out one's own, as against surrendering to conformity with a model imposed on us from outside, by society, or the previous generation, or religious or political authority" (Taylor 2007, p. 475). This is also an age of public and mutual 'display' where fashions, commodities, and so on become opportunities and vehicles of individual expression and 'the self-definition of identity'—even if that identity is not so much autonomous as linked to a broader cultural identity shared by perhaps millions of others (Taylor 2007, pp. 481–83).

Before progressing, it will be useful to pause for a moment and consider what we might learn from these reflections on the question of personal identity. First, if the concept of identity is complex, it is also contested, both with respect to its development and structure, and also in terms of its significance for self-understanding, which might in some cases be a healthy and integrating sense in a person's life, or in other cases may become a personally isolating or socially divisive feature. Crawford and Rossiter highlight the dynamic and necessary balance between internal/personal and external/cultural resources required for mature and healthy identity, as well as the moral contours of an identity. They insist that identity formation includes the passing on of content as well as a firm subjective appropriation of one's identity. SIT reminds us that personal identity is inevitably social and often complex, in accordance with the disparate groups to which one belongs. It provides practical insight into the ways group participation contributes to identity formation, especially when persons move between groups or enter new groups. It is noteworthy that SIT has achieved some prominence in recent contributions to biblical scholarship (see, for example, Tucker and Baker 2014; Kok 2014; Du Toit 2023; Ukwuegbu 2008). Fukuyama, Appiah, and Grant are concerned that contemporary discussion of identity has become reductive and thus detrimental to common life. From Fukuyama, we learn not only that the quest for identity is a soul-cry for recognition, but also that

the secularization of the concept in modernity has made this quest a social and political imperative that displaces other rational and moral dimensions of human social existence. Appiah reminds us that one's identity is communal, historically conditioned, multiple, and that the various aspects of a person's identity are mutually conditioning. Grant warns that valorising a particular or singular identity can lead to tyranny, and to the exclusion of genuine freedom and love, and warns in a manner similar to Appiah that any concept or practice of identity which serves to isolate us from others or sets us over against others is somehow deficient. Finally, Taylor has identified what he has termed the three transcendental conditions of personal identity, and the central significance of allegiance to a transcendental Good which orients, and provides direction for, one's life. Identity is framed in terms of a worldview and value system, a social imaginary which is sometimes held unconsciously or unreflectively but supports a stronger sense of identity when articulated.

### 3. Baptism and Identity

In light of these reflections, then, what might we say of Christian baptism?[9] The first thing to note is that lessons drawn from these reflections are not necessarily alien to the theology and practice of Christian baptism. But second, and to the contrary, we must also concede that this same theology and practice also cuts across modern conceptions of identity in some decisive and possibly irreconcilable ways.

To begin, perhaps the simplest idea is Fukuyama's contention that the quest for identity concerns an innate soul-cry on the part of every person for recognition, acknowledgement, and worth. Christian baptism speaks profoundly to the individual at the level of personal identity. One need not, for example, take 1 Peter as an ancient baptismal liturgy or catechetical tract,[10] to appreciate its various references to washing and baptism, as these are set within an overarching narrative of the electing love of a God who has chosen and washed his people, made them his children by new birth, granted them the living hope of an eternal and imperishable inheritance, redeemed them with a price of infinitely more value than silver and gold, and dignified them by naming them with the ancient covenantal titles originally applied to Israel, including becoming a royal and holy priesthood in his temple (see 1 Peter 1:1–2:10).

So, too, baptism into the name of the Father, and the Son, and the Holy Spirit (Matthew 28:19) places our "unique and personal name in the company of the Trinity. . . . Holy baptism redefines our lives in Trinitarian terms" (Peterson 2005, p. 303). According to Eugene Peterson, this focal practice is basic to Christian identity formation, to our new identity as a child of God. While Thomas Oden's claim that trinitarian doctrine and theology emerged from the baptismal practices and liturgies of the ancient church is cogent, far more than doctrine is at stake here (see Oden 1992, p. 12). Baptism is an entrée into a participation in the life of the triune God, into the eternal communion of love, peace, fellowship, and mission that is the triune life, and into an experience in which the baptizand is now embraced, as they find themselves loved and accepted, acknowledged and recognised, forgiven and reconciled, by a love that posits and creates worth.[11] Further, they are also inducted as it were into the life of *this* people, the community formed in common baptism, and among whom too they are received, acknowledged, and valued.

These reflections also address the matters raised by Taylor, affirming his insistence that personal identity is grounded narratively, though doing so in a manner that cuts across the 'great disembedding' or 'disenchantment' that characterizes modern secularity (see Smith 2014, pp. 27–30, 45). Taylor correctly notes that the modern social imaginary constitutes a worldview and value system that is "largely unstructured and inarticulate", an understanding "that can never be expressed in the form of explicit doctrines" (Taylor 2007, p. 173). This is not the case with respect to Christian baptism into the triune name which constitutes an explicit *re-enchantment* or *re-embedding* of the life of the Christian and the Christian community in an alternative 'social imaginary' grounded in and emerging from Barth's 'strange new world of the Bible', and understood precisely in terms of a robust theological realism.[12] This is to say that baptism involves a thorough-going conversion of

the imagination, though the new world into which we are baptised is not imaginary. It is rather the world of God, Father, Son, and Holy Spirit, the cosmos created and loved by God and reconciled in the Son and intended by God for glorious eschatological renewal through the power of the Holy Spirit. The community of the baptised are those who through baptism have been initiated into this world, into a new relationship, an ongoing story the unfolding of which has been told for millennia, and a living and traditioned community in and among whom they find that they have been granted a new identity. The story of the individual believer has been caught up into the larger and encompassing story of God and of God's relation to his people and the whole created order more generally. Their individual narrative has been resituated, their personal story re-narrated through an entirely new set of lenses. The Christian no longer lives in a world of self-sufficing humanism, and no longer is the highest value and allegiance of life that of their own human flourishing. Rather, the chief end of human existence is to know and love God, and to enjoy him forever.[13]

Christian baptism also cuts across the foundations of modern identity philosophy in at least three additional ways. First, there is something profoundly illiberal about it. Christian baptism is not a designer-project curated for an Instagram profile, not merely one lifestyle decision in a series of such choices, or something one can 'experience' in the present before moving on to other, newer, more enlivening experiences elsewhere. No one can baptize themselves. Baptism is an act of the Christian community; one is baptised *by* the church and *into* the church (Witherington 2007, p. 118). As such, it involves a certain yielding up and surrender of the self, to the process of baptism, and to the church, as well as to Christ, who commanded the baptism, and the triune God into whose name one is being baptised. In an act of personal sovereignty, one surrenders the seat of that sovereignty to become one under the command of, and obedient to, another. That is, obedient to the one Lord of the church in the company of the church.

Baptism is not merely an act of the church, however, but as Wolfhart Pannenberg argues, it is also an act of God:

> Administering baptism is indeed a human act, but at its core it is a divine action on the candidates. For to be baptized in the name of God is to be baptized not by others but by God himself, so that even though others administer it, it is truly God's own work. (Pannenberg 1998, pp. 260–61)

In the act of baptism, God lays claim to the person's life, definitively linking the life of the person baptised to the destiny of Jesus (Pannenberg 1998, p. 260), implanting their existence in Jesus Christ (Pannenberg 1998, p. 237), such that their very personhood is now reconstituted by this relation to God, and "concretely by participation in the filial relation of Jesus to the Father". As such, says Pannenberg, baptism is "the constitution of Christian identity" (Pannenberg 1998, p. 239). In this way, too, baptism resists the modern romantic ideal of a self-grounded and self-defined identity.

Second, this means that Christian baptism involves a radical decentering of the self and its earthly identity-markers. In Christian faith, one's identity is a divine gift that comes to us from without, but which is also wholly self-involving, claiming the recipient in the totality of their existence as an existence for another. Galatians 3:27–29 indicates that those baptised into Christ have 'clothed themselves with Christ' and so now 'belong to Christ' such that, in the new community, there is "neither Jew nor Greek, slave nor free, male nor female: for you are all one in Christ Jesus" (see O'Neil 2019, pp. 13–22). Although we must surely insist that the order of redemption does not dissolve the order of creation—a male Jew or a gentile woman retain both their gender and their ethnicity—we might also insist that these identity-markers are displaced, secondary, and henceforth, subordinate (see Snodgrass 2011b, pp. 268–69, 272; and Dueck 2011, p. 26). The baptised have died with Christ with the result that their life now is "hidden with Christ in God" (Colossians 3:3).[14] And—here is the real challenge—in the new community, truly equal recognition and dignity is accorded to all the baptised regardless of their gender, ethnicity, or socio-economic status. In baptism, a new polis is constituted, and one becomes the member of a new and different family

(Smith 2009, pp. 182–87). A re-ordering of identity takes place, and with it a transformation of social and relational priorities, customs, and practices.

Not everyone concurs with the position taken here, and the matter is a live discussion in Christian scholarship and missiology. Some scholars, for example, insist to the contrary that to speak of the relation of Christian and cultural identity in terms of 'primary and secondary' is misleading and unnecessary, that there is no need to pit these two aspects of identity against one another, for they may be construed as existing on a continuum. It is certainly true that disciples of Jesus are not required to leave their cultures to follow him, and that their manner of discipleship will reflect certain aspects of their distinctive cultures. Nevertheless, Christians in every culture and age will face again and again the call to Christian faithfulness in ways that challenge their cultural mores, convictions, and heritage. That Christians inevitably struggle with this call to express their faithfulness appropriately is evident to anyone who examines (for instance) modern western Christianity.[15] Roccas and Brewer's fourfold typology does duty here. To live only in accordance with one's inter-sectional identity ("I can associate only with other English, female, Cambridge-educated lawyers who are also Christian, tennis-playing mothers") or to compartmentalise one's representation ("I will live and act as a Christian when amongst Christians but not when in other contexts") is clearly untenable for Christians. It is more difficult to parse the better option between 'dominant' and 'merged' models of complex identity. On the one hand, the merged model of identity complexity recognises the legitimacy and complex interplay of one's multiple identities and so potentially increases tolerance of others, in acknowledgement that they are not solely this identity ('person') or that, but both—and more—simultaneously. As Kok noted twice in his essay, when discussing Paul's ability to transcend social boundaries and facilitate a higher level of inclusivity: "This is particularly inspiring, even today" (Kok 2014, p. 8).

In this mode, ingroup identification is extended to others who share any of one's important social category memberships. Thus, the merger model goes beyond additivity of multiple ingroup memberships to what Urban and Miller (1998) referred to as the 'equivalence pattern' of evaluating others with multiple group memberships. The more social identities the individual has, the more inclusive the definition of ingroup becomes, to the point where no sharp ingroup–outgroup distinctions are made on any dimension and all others are evaluated equivalently ((Roccas and Brewer 2002, p. 91), referencing (Urban and Miller 1998)).

Roccas and Brewer would evaluate all identity ascriptions equivalently, rendering Christian identity as merely one identity alongside other identities, and thus blurring or even obliterating all ingroup–outgroup distinctions. The mechanism accomplishing this is a primary commitment to a hidden, otherwise unidentified 'superordinate principle', one which in all likelihood is culturally derived and self-chosen. In effect, this becomes an a priori identity commitment to which all other identity ascriptions are subordinated. The problem here is evident. It would appear that the only 'superordinate principle' that may legitimately be introduced in a Christian discussion of identity complexity is Jesus Christ himself, as he is witnessed in the Scriptures as paradigmatic identity prototype. Christian identity becomes the overarching identity ascription within which other identities may be recognised—or rejected. Thus, in the example noted earlier, the Thai-Christian now living and working in the United States has effectively subordinated their Christian identity to that of 'American'. This, too, is untenable for Christians who would be faithful to the claim of the New Testament.

Although Kok argues that the merger model may be fruitful in New Testament studies, he seems alert to the dilemma:

> Practically, in Paul's day and in his mind it meant that the old categories of distinguishing between insiders and outsiders should be drawn in a new inclusive way by means of an overarching identity *in Christ*. (Kok 2014, p. 8, original emphasis)

Kok's identification of the superordinate principle is crucial. Only in Christ may Christians have a properly theological measure by which to evaluate competing identity

claims. In Christ, those of various ethnicities, socio-economic and cultural backgrounds, whether male or female, may be recognised, valued, and affirmed. Nevertheless, some identity ascriptions may be regarded as incompatible with life in Christ and thus incapable of equivalence. The history of the church provides examples. In the patristic period, Christians wrestled with whether soldiers and others whose vocation involved killing people could retain this vocation and yet be a Christian. In more recent centuries, Christians finally rejected the idea that one could be a Christian slave-trader. That is to say, Christian faith generates norms which provide direction and standards against which other identity claims are tested. As we will note presently, this does not require Christians to maintain hard-edged and rigid boundaries between insiders and outsiders, although it does mean that certain identity claims are incompatible with life in Christ. When Christian identity becomes the superordinate principle whereby all other identity claims are evaluated and thereby affirmed or rejected, it appears that in fact, Roccas and Brewer's dominance model more closely corresponds with the function of Christian identity vis-à-vis other identity claims.[16] To put it otherwise, Christian identity aims for salience in one's self-concept.

Finally, against the Romantic notion of the inherent goodness of the human personality, water baptism as *washing* (see, for example, Acts 22:16 and Hebrews 10:22) reminds us that, from a theological perspective, there is at the heart of human identity and personality a profound brokenness or stubbornness or waywardness, a narcissistic pride that insists on self-assertion and self-expression over and against God and against others. The contemporary insistence that we discover and express our 'true identity' hidden in the depths of our solitary being breeds an anxious self-concern. When a person exalts the self as the true locus of identity and value, their relationships with others are disturbed and ordinary human desires are distorted (Cooper 2003, p. 57). Such exaltation constitutes an idolatry of the self, together with a corresponding attempt at the justification of one's choices. Water baptism, and the repentance commonly associated with it (Acts 2:38), challenges this easy acceptance of our own goodness, confronts us with the reality that our identity is in fact 'mixed', and yet also reassures us that our sins can be forgiven, that we can be cleansed, that we are in fact loved and accepted, and even liberated from the requirement of establishing our own worth.[17]

### Particularity and Openness

I have argued thus far that baptism is constitutive of a Christian's identity. I have tried to do so in conversation with a few insights drawn from the work of scholars examined above, and showing some of the ways in which Christian baptism both resonates with and cuts across these reflections. Now I want to take up Stan Grant's warning with respect to the dangers of a totalising identity. Christian baptism has to do with Christian particularity, with a new identity in which some previous forms of life are renounced and left behind and a new way of faith and discipleship in relation to Jesus Christ is established (Mikoski 2009, p. 205). Yet, as Gordon Mikoski argues, particularity must not be set against a fundamental *openness* toward the world. Both particularity and openness must be our goal (Mikoski 2009, p. 201). If Christian identity becomes an end in itself, a means of sectarian withdrawal from the world in all its vibrancy, richness, need, and depravity, if the pursuit of Christian identity leads the Christian community into supposed enclaves of holiness where they are protected as it were from the evils of society and culture, they are well on the road to falling into the kind of petty tyranny that Grant rails against. Further, they have also, in truth, denied the reality and meaning of their baptism.

I have already noted that baptism into the triune name involves a participation in the triune life of love, peace, fellowship, and *mission*. To be baptized into the name includes being caught up in the divine mission of healing and reconciliation exemplified in the life and ministry of Jesus and committed to the church in his name. The divine being, revealed in the trinitarian portrayal of God in Scripture, is that of an ecstatic outpouring of divine self-giving and hospitable love. In the suffering redemptive love revealed at the cross, the Son of God embraced the misery and death of all humanity, and in so doing demonstrated

the openness that characterises God's relation to the world. "God was in Christ reconciling the world to himself, not counting their trespasses against them; and he has committed to us the word of reconciliation" (2 Cor. 5:19).

Karl Barth argues that Christian baptism corresponds to and finds its basis in the baptism of Jesus Christ (Barth 1969, pp. 54–61). In his own baptism, Jesus responded to the word addressed to Israel in the proclamation of John and offered himself in unreserved submission to the will of God. At the same time, his baptism was also an offering of himself in unreserved solidarity with humanity—with humanity in its alienation and distance from God, in its sinfulness and misery, its longing and hope. Thus, in his baptism, Jesus committed himself unreservedly to the service of both God and humanity and so entered his ministry as the mediator between God and humanity. Just as Jesus was baptised into a deepening solidarity with humanity which was as such his service to God, so Christian baptism is a baptism into the common ministry committed to the church. The person baptised is

> now personally co-responsible for the execution of the missionary command which constitutes the community, of the commission to the outside world which surrounds both it and him on a large scale and a small scale alike. The task of every Christian—not additionally but from the very outset, on every step of the way assigned to him in baptism—is the task as a bearer of the Gospel to the others who still stand without. ... The baptism from which he comes was as such a consecration or ordination to take part in the mission which is committed to the whole Church.[18]

Finally, the dialectic of particularity and openness should also characterise relationships within the Christian community, lest the church also become a dominating and oppressive agent with respect to her members. Nor may the church be so rigid or arrogant as to assume that it has already apprehended the eschatological fulness of truth and life. Believers are baptised *by* the church and *into* the church but not *to* the church to become, as it were, the possession *of* the church. To be baptised into the triune name is to come, says Frederick Bruner, "onto the account and into the possession of the Great God; baptised believers come under new management. They are transferred to a new company" (Bruner 2004, p. 821). Or as Pannenberg has reminded us, God is acting in and through the church, claiming us for himself. As such, the church baptises new believers not *to* itself but in company with itself *to* and *for God*. Certainly, this will include distinctively Christian notions of what baptism signifies, of the identity and life appropriate for those baptised since, as Appiah has noted, identities are communally grounded and historically conditioned. Christian identity comes with very definite 'entanglements', some of which cannot be compromised; some forms of belief or action are incompatible with Christian identity. Other 'entanglements' may require, again as Appiah suggests, some interpretation and negotiation (Appiah 2018, p. 217). The reason for this is that in baptism, one's life is 'set in direction of the Good', as Taylor says, a firm, settled direction and commitment in which we are *becoming* or *moving* or *going* (Taylor 1989, pp. 46–47). That is, a space is opened here for variety of expression, for conformity *and* liberty, for fresh initiatives and new insights as believers—within the ongoing life, theological reflection, and mission of the Christian community—oriented and responsive to the one Lord of the church. Christian formation does not occur in a cookie-cutter fashion, even though it will have some clearly discernible processes, forms, and results. Rather, it sets believers into the relational company of the triune God and the people of God amid an ongoing and unfolding life with God.

Thus far, I have explored various interpretations of the concept of identity in late modern western contexts, and brought these interpretations into dialogue with the Christian practice of baptism. This dialogue has proven fruitful, identifying a degree of resonance between baptism and contemporary identitarian philosophy, some thoughtful considerations for the church's practice, as well as identifying some aspects of Christian baptismal practice which resist assimilation to contemporary thought and mores. Some of the issues arising deserve further careful consideration by churches as they approach the task of

formation of baptismal candidates. First, and perhaps most significant, is the observation that mature identity and healthy identity formation requires subjective appropriation and is not merely the passive acceptance of an externally imposed view of the self. A person chooses their identity and increasingly identifies with it in their self-understanding and expression. Because identity is also plural, Christian identity formation will press for primacy in one's self-understanding, without suppressing identity complexity. Second, however, is the reality that identity is formed dialogically, utilising both internal and external resources, finding a balance between personal and communal inputs and standards. This requires parsing the relationship between personal and ecclesial identity such that the individual freely embraces a communal identity as their own, conforming themselves, however imperfectly, to an external criterion—Jesus Christ—in company with others. This is all the more challenging in an age in which such conformity is deemed inauthentic. Finally, Crawford and Rossiter insist that healthy identity formation involves both process and content, while Taylor reminds us that one's identity is inescapably moral, involving an orientation to the Good. Christian faith and theology provide distinctive content as to the nature of this Good—the triune God revealed in the history of Jesus Christ—as well as the contours and content of the moral life. Some indications of how churches might practically approach the matter of baptismal formation is the focus of the final section of this essay.

### 4. Baptism and Formation

I begin with an assertion: Baptismal identity is both a divine gift and an ecclesial task.[19] Bordeyne and Morrill are surely correct in their contention that there can be no 'baptismal positivism': "any notion that a rite such as baptism could singularly, unequivocally govern the thoughts, imaginations, and actions of any and every Christian participating in it" (Bordeyne and Morrill 2012, p. 158). The question that arises here can be put bluntly: Is there a 'grace of baptism'? Is God active in baptism or is it purely a work of human response to the reception of saving grace received otherwise through the proclamation of the gospel and the hearing of faith? Gordon Fee, for example, argues that baptism is the human response to the Spirit's prior work of conviction, the 'hearing of faith', regeneration, and empowerment in which the believer offers himself back to God for life and service in his community (Fee 1991, p. 117; See also Ladd 1993, pp. 587–88, 593–94). He views experience of the Spirit as the *sine qua non* of Christian conversion, and argues that in Paul, the gift of the Spirit is not associated with baptism.[20]

Other readers of Scripture, however, note the close association of the Holy Spirit with baptism, even in Paul.[21] Including a broader range of biblical texts provides additional ground for viewing baptism as a 'means of grace', while even Dunn admits that sacramental interpretations of Paul have a strong exegetical basis.[22] This is observed especially in relation to Paul's references to baptism in Romans 6:3–4, Galatians 3:26–28, and Colossians 2:11–12, in which the accent is not on the believer's reception of the Spirit but their union with Christ in his death and resurrection. These texts, it seems to me, suggest an intrinsic relation between baptism and this grace. Indeed, with respect to the Colossians passage, Paul's entire argument in 2:8–3:17 seems predicated on a realist account of baptism. Finally, the New Testament portrays a multi-faceted cluster of divine and human actions which together comprise conversion–initiation, although theologians invariably order them in different ways. These actions include the conviction of sin and regeneration by the Holy Spirit, repentance and faith in response to the gospel, water baptism, reception of the Holy Spirit into one's life, often in a dynamic and even visible experience, and incorporation into the life and ministry of the Christian community.[23] Dunn thinks it likely that Paul viewed this cluster as a 'complex whole', with baptism filling "an important role within the complex whole", even as "the moment and context in which it all came together" (Dunn 1998, pp. 455, 457). Without denying that baptism does have the character of a human pledge in grateful response to grace, and that unless the human act is 'mixed with faith', it may indeed prove vain, it is nonetheless an event in which divine and human action coincide, and grace is given as well as received.

Thus, while I concur that 'baptismal positivism' is undesirable and impossible, it remains important for churches to affirm that baptism is not simply an 'empty' rite, and to encourage congregations and candidates to approach it reverently, prayerfully, and expectantly. Churches ought to pray fervently with and for baptismal candidates—perhaps also with fasting and the laying of hands (see, for example, Kreider 2016, pp. 145, 182; Lane 2020, p. 102; O'Loughlin 2011, pp. 81, 92n17, Cf. *Didache* 7.4)—that they might be granted and experience manifestations of divine grace as part of their baptism. Such formative practices involve the whole congregation and provide gestures and context which deepen faith and heighten anticipation for the divine work.

Nor should churches shrink from the possibility of baptismal experience but rather be open to it, though without manipulation or coercion. Paul apparently considered the experience of the love and power of God essential for Christian faith (1 Cor. 2:4–5; cf. Gal. 3:2–5; 1 Thess. 1:5–6), especially if it is to be sustained in times of suffering and trial (Rom. 5:3–5). The assurance of faith is necessary for every Christian, and much can be gained through the grace given us in our experience of conversion–initiation in which, as we have seen, baptism plays a central and integrating role. Churches do well to make water baptism the central ritual in which the entire cluster of saving actions noted above are discussed and learned, tested, affirmed, encouraged, prayed for, and testified—in hope that candidates are decisively embedded in their new life, identity, and community, in the grace of our Lord Jesus Christ, and the love of God, via a life-transforming experience of the mighty power, presence, and fellowship of the Holy Spirit.

Bordeyne and Morrill's rejection of 'liturgical positivism' recognises the broader context of the baptismal rite and thus also the ecclesial task of Christian identity formation—the purposeful inculcation of subjective realisation of Christian identity by baptismal candidates. The New Testament, they suggest, makes it abundantly clear that baptism does not settle matters for the believer concerning who they are and what they should do, but frames them (Bordeyne and Morrill 2012, pp. 158–159; cf. 164). The ritual does not stand on its own but within the broader context of the gospel, the community, the call to discipleship, the church's mission, and so on. It requires "an ensemble of references, of stories, of practices, of visions of the world in order to shape and reshape the identity of the disciples of Christ, in dynamic relationship with ethical behavior and its spiritual importance" (Bordeyne and Morrill 2012, p. 166). Although this 'ensemble' will feature a variety of instruments in different contexts, the following common elements are central to the framing work of baptismal identity: induction into a new story, a new community, and onto a new path.

First, baptismal candidates must learn—in such a way that they begin to inhabit—the 'Big Story', the master narrative of God and his people. This is the biblical account of God as Creator and Redeemer, in all his dealings with his creatures from creation to consummation, including his covenants, promises, judgements, and purpose. The account especially focuses on the coming of Jesus Christ as 'God with us' (Matt. 1:23), his life and ministry, suffering, death, resurrection, and ascension, the outpouring of the Holy Spirit, and the revelation of God's universal and cosmic purposes in and through the church as we 'wait for and hasten' the return of Christ and the restoration of all things in a new heaven and a new earth (2 Pet. 3:12–13; Acts 3:21). The Big Story is a revelation of the Good—the eternal love and purposes of God revealed in Scripture and supremely in Jesus Christ—which, as such, grounds and portrays a distinct view of the reality within which we have our existence, and also constitutes the identity of the people of God. As Taylor emphasised, we grasp our identity in a narrative that helps us make sense of our lives and orients our life toward the Good, setting us in the direction of the Good on a path in which we are not merely existing but *moving* and *becoming* (Taylor 1989, pp. 41–48). The Big Story provides the worldview and value system, the social imaginary into which the baptizand is being inducted, the 'strange new world of the Bible' they now call home. It includes also the history of the church and the particular community of which the new believer is now a part. Churches might inculcate a '*shared awareness of the present Christian community as the primitive community and the eschatological community*' (McClendon 2002, p. 30, original emphasis),

that congregants might find in this (unfinished) story their own story and calling, and learn that God's purposes continue to be prosecuted and realised in and through this people. They are set in the company of Moses and Mary, Huldah and Haggai, Paul and Priscilla. The waters of their baptism are those of the Red Sea and the Jordan, and of the myriads of Christians who through the centuries have also walked this path. What God is up to in their lives is a continuation of what God has been doing over many centuries—liberating, sanctifying, transforming (2 Cor. 3:18) (see Dueck 2011, pp. 21, 25).

Thomas O'Loughlin has written that Christian formation is "calculated to irreversibly alter the habits of perception and standards of judgment of novices coming out of a pagan life style".[24] Thus, the aim of being inducted into this story is the 'renewal of the mind' toward a transformed life (Rom. 12:2). This is more than the mere accumulation of biblical knowledge, important as that is. It seeks also the conversion of the imagination to see and inhabit the world anew, in light of the theological account provided in Scripture; the alignment of the human will to God's will; the tethering of the affections in humility and trust to the love and fear of God; and the development of Christian modes of theological reflection and moral deliberation. This will involve an array of strategies when reading Scripture—historical, devotional, moral, missional, etc.—always with the intent of believers hearing the Word within the word, in dialogue with the community of faith past and present, and integrating it into their lives personally and corporately, not just learning it intellectually, but learning to practise and embody it in their lives together.[25]

Second, believers are incorporated into the new community of God's people, a counter-cultural people intended as the 'salt of the earth, the light of the world, and a city on a hill' (Matt. 5:13–16). To be baptised is to be plunged into Christ's body and united with one's fellow believers. One becomes a participant of its life and ministry, a partner in its joys, suffering, and hope, and a recipient of God's grace, blessings, and promise given to his church. The corollary at the local level is that the baptised person is welcomed into the community as a brother or sister in Christ, accorded the status and honour worthy of a child of God, regardless of creaturely or social distinctions, recognised in time for the spiritual gifts and ministries given them, and their contribution to the fellowship nurtured and valued. It matters a great deal that the believer is truly loved in this new community, that they are embraced in relationships of acceptance, care, mutuality, and friendship that replicate in human form the high-tensile, tender-hearted love of God. Although this is especially true for those who come to faith with addictions, behavioural disorders, or wounded hearts, every Christian harbours secret sins, weaknesses, hurts, and faults. In such cases, transformation requires the church as a community

> of loving people who bring the healing grace of God to bear on the life of the disordered person. A long-term, deep-reaching transformation of the broken heart—the kind that frees the bound human will and strengthens the weak human will to choose righteously—is most often cultivated over time through relationships that finally convince the person of God's love. (Thigpen 1992, p. 51; See also Thrall and McNicol 2010, pp. 61–83)

Only where a person knows that they are deeply loved will they risk the vulnerability of exposing their wounded and wayward hearts to the healing light of God's love and truth. Personal and permanent transformation requires personal relationships (Clark 2003, pp. 252, 256).

The identity-forming recognition grounded solely on one's status in Christ cuts two ways. It affirms and elevates those who previously were without status and recognition but will challenge the identity of new believers if their self-concept is grounded in some creaturely or cultural standard. The rich person in James 1:9–11, for example, is forbidden henceforth from boasting in their riches, the 'glory' of their face, or any other benefit that may have previously accrued to him or her.[26] Rather, they are to embrace their (social) humiliation in light of the new theological construal of reality that they have obtained: knowledge of eschatological judgement and the standards there applied. Baptismal identity has social implications in the new community, as Klyne Snodgrass has argued. Although

racial, gender, or socio-economic distinctions may still exist in the community, valuation based on these distinctions may not (Snodgrass 2011b, p. 268). These realities will likely continue to determine identity, but can no longer have primary defining force; they must be made subservient to the gospel and to Jesus, who now as Lord has been given primary defining force in one's life (Snodgrass 2011b, pp. 272–73). Baptised into Christ's death and resurrection means that the new believer has been crucified with Christ and thereby 'displaced' from their own being. "If they are not willing for this to happen", says Snodgrass, "he or she cannot become a Christian" (Snodgrass 2011b, p. 264). Snodgrass's rhetoric may overstate the matter given that we grow into our Christian identity over time. Nevertheless, how hard it is for those who cling to their socio-cultural identities to enter the kingdom of heaven!

Because Christian identity is mediated via the community, more 'caught than taught', it is imperative that the community into which persons are baptised be living out its calling and identity faithfully. In a study that explored why many who considered themselves Christians in their young adult years subsequently withdrew from evangelical faith, Steven Garber writes that

> Over the course of hours of listening to people who still believe in the vision of a coherent faith, one that meaningfully connects personal disciplines with public duties, again and again I saw that they were people (1) who had formed a worldview sufficient for the challenges of the modern world, (2) who had found a teacher who incarnated that worldview and (3) who had forged friendships with folk whose common life was embedded in that worldview. There were no exceptions. (Garber 1996, p. 111)

This sobering finding shows that learning and understanding the Big Story is not sufficient on its own but must be supplemented with a community in which the worldview, virtues, and values of the Story are embedded and embodied. Mature Christian faith and identity requires congruence between one's inner experience of their relationship with God, a personal ideal grounded in a coherent theological vision, and their public presentation. But it is difficult if not impossible to achieve this in isolation. Thoughtful participation in common worship, spiritual conversation, and corporate theological reflection can help overcome a privatised faith and encourage authentic spiritual and moral formation (Clark 2003, pp. 252–56). Also critical, as observed by Garber, is the friendship of a teacher or mentor who incarnates the way and ethos of the Christian worldview. New Christians need role models and exemplars who embody the Christian life and thus show both its possibility and how it translates in practical terms in the context. SIT researchers, in particular, have emphasised the importance of the group's leaders embodying the vision and norms of the group: "As the (most) prototypical group member the leader best epitomizes (in the dual sense of both defining and being defined by) the social category of which he or she is a member" (Esler 2014, p. 34). The mentor accompanies the novice, becoming a friend and confidant, praying for and with them, watching their progress and struggles, supporting, helping, and encouraging them to live an authentic Christian life. In community, new believers learn to be Christians by learning to do what the Christians do, in their worship and prayer, service, mission, care, relationships, and interactions. They learn by seeing, by participating, by learning the practices of the new community as it, too, lives a life of discipleship to Jesus Christ (see Kreider 2016, pp. 156–60).

This all suggests that third, new Christians must be inducted into 'The Way' (Acts 9:2; 19:9, 23), that is, set in a new direction on a path different to the path they previously travelled. This is not an idiosyncratic or self-chosen path but that which corresponds to and continues the Big Story in the life of the new community of which they are now a member. Baptism, particularly by immersion, vividly portrays the death of the old self and its ways and the being raised to a new life in Christ (Rom. 6:3–7; Gal. 3:27; Col. 3:9–11). The Christian has died to their former life and its identity ascriptions (Gal. 2:20). Crucified with Christ, it is no longer 'I' that lives but Christ that lives in me. Christian identity is that of one who has been crucified, whose self has been displaced and who henceforth lives

from a different centre—Jesus Christ. To live by faith in the Son of God is the act of living out the identity God gives us by grace (Snodgrass 2011b, pp. 262, 264–65).

The Christian life, therefore, is a matter of 'learning Christ' (Eph. 4:20–24), which involves practical embodied adoption of the teachings of Jesus as one's rule of life. The requirement to 'teach them to obey all I have commanded you', together with baptism, constitutes the life of discipleship appropriate to the Great Commission (Matt. 28:19). Christians are those who, together as one body in common life, live the way of Jesus Christ. The emphasis is on formation in obedience, not merely in knowledge, on disciples who in practice have been 'fully trained' so that they are 'like their master' (Luke 6:40). O'Loughlin likens Christian discipleship to an apprenticeship, to learning a craft from a master craftsman.[27] This is a relational model of identity formation, in which the mentor has so internalised the Way that they are able to live it reflexively and thus also to model it to the one they are accompanying. Apprentices hopefully will recognise in their mentor standards of Christian excellence to be emulated and will thus give themselves to the required disciplines of the training which is, as Eugene Peterson has written, 'a long obedience in the same direction'.[28] In his study of the *Didache*, O'Loughlin notes that the content of the Way amongst the earliest Christians was primarily the distinctive teaching of Jesus, drawn especially from the Sermon on the Mount, supplemented by the Decalogue, and the patterns, practices, and routines of life in the community. One was typically drawn to the message of Christ through relationship with someone who was already a Christian. In this relationship, they would be confronted with a fundamental choice to accept the way of life and reject the way of death. That step taken, they could be baptised, the actual baptising being done by the church member who had trained the newcomer in the Way.[29] It was by engaging in a relationship with a Christian and entering into the Christian Way of life that one became a disciple. One became a Christian by doing what Christians did. The *Didache* assumes that "the neophyte will only really know discipleship from the inside, living it" (O'Loughlin 2011, p. 84).

These three central elements of Christian formation frame the subjective appropriation of Christian identity. It is evident that they include elements of both process and content, as Crawford and Rossiter have insisted. Is there an overarching process by which this formation might be consistently applied in Christian ministry?

Alan Kreider has argued strenuously that contemporary Christian formation should be shaped by the practice of the ancient church's catechumenate—an argument that unites both goal and process. He identifies a new *habitus* as the aim of this formation: reflexive behaviour that corresponds to the central dispositions and lifestyle of the Christian community's common life (see Kreider 2016, pp. 133–84; here: 165–66; See also Kreider 2011). The Christian *habitus* was a way of life rooted in a host of biblical passages and especially in the teachings of Jesus—that over time and with practice, became embodied and habitual (Kreider 2016, pp. 165–66).

> How did Roman Christians become habituated so that they lived the way of Christ reflexively? . . . Was it possible for a community to develop the practices necessary to maintain its new life over against a pagan habitus that was well established and deeply seductive, respected by society's elite, informed by deep narratives, and made immediate by omnipresent visual arts? In such a situation, could the Christians physically renounce the old habitus and supplant it with a new habitus? And if so, how? (Kreider 2016, p. 144)

Kreider argues for a renewal of the ancient catechumenate as a process for formation in the contemporary church, illustrating his contention with reference to the *Didache*, Justin Martyr, Cyprian, and the *Apostolic Constitutions*. He claims that the extraordinary growth of the pre-Constantinian church was due in large part to the 'uncommon commitment' of the ancient church to this formation (Kreider 2016, p. 2). Catechumens were to learn and embody the way of Jesus as a pre-requisite for baptism, as a 'counter-habitus' to the culture and ethos of the Roman empire (Kreider 2016, p. 143). Conversion involved the re-formation of the new believer's identity and life "in which the candidate declares

that Jesus is Lord, identifies primarily with the Christian family ("I am a Christian"), and commits himself or herself to living in the Christian way" (Kreider 2016, p. 176).

Although this and similar proposals have attracted their critics, not least as an offence to a doctrine of grace (Colwell 2005, pp. 131–32; cf. Witherington 2007, pp. 125–26; Lane 2020, pp. 99–100), W. John Carswell laments the lack of "an intentional and cohesive *process* of initiation and inculturation" to enable candidates to live *into* the baptised life (Carswell 2018, p. 431 (original emphasis)). Although a member of the Church of Scotland, he commends the Roman Catholic *Rite of Christian Initiation of Adults* (RCIA) as a model for the Reformed Church, precisely because it updates the ancient catechumenate, which

> Prepared its baptizands for a radical departure from a hostile and pagan culture . . . baptism meant leaving one community and joining another, a transition marked by great ceremony and serious personal reflection from the participants. Catechesis was lengthy and challenging and called for genuine life change. (Carswell 2018, p. 434). See (International Commission 1987)

## 5. Conclusions

It will be clear, I think, that I have undertaken this exploration of identity and baptism as one who practises believer's baptism, and so with the assumption that the baptismal candidate is of an age to receive instruction concerning what baptism signifies and requires, and to participate in the Christian community and its various formative activities and practices. This is not to suggest that the study is not also relevant to those who practise infant baptism, although with structures and practices appropriate for formation in those traditions. Gilbert Meilaender, for example, grounds his study of Christian bioethics in the reality of our identity established in (infant) baptism:

> In baptism we are handed over to God and become members of the Body of Christ. . . . In baptism God sets his hand upon us, calls us by name, and thereby establishes our uniquely individual identity and destiny. We belong, to the whole extent of our being, only to God, whom we must learn to love even more than we love father or mother. (Meilaender 2013, p. 2)

In the story of her conversion from a pagan life to Christian faith, Rosaria Champagne Butterfield notes that the pastor of the Reformed Presbyterian Church who was guiding her was concerned to ensure that she had never repudiated her baptism into the Catholic Church. Her conversion did not include a 'rebaptism' as may have occurred if she were converted in a Baptist context, but it did include an acceptance of the validity and providential grace involved in her infant baptism. Further, her conversion included official vows—a 'Covenant of Church Membership'—which functioned in this church as a reaffirmation of baptismal commitments.[30]

In both cases, whether adult or infant baptism, the subjective appropriation and expression of a truly Christian identity is necessary and requires formation. I have argued that such identity formation is both a gift and a task, a process in which the candidate is inducted into the Big Story, the Christian community, and the Way of Jesus, such that their identity becomes simply and primarily, "I am a Christian". This is a matter of 'slow conversion', a deepening comprehension and grasp of the grace of Christ in our lives, as well as a deepening commitment to live in the way of Christ in the company of his people, in mission and in hope. The role of baptism understood as both the rite and the broader context of conversion–initiation is integral in this formation.

I began this essay with the story of Polycarp, whose martyrdom set his Christian identity in stark relief. I close with a similar story, this time of a modern martyr—Dietrich Bonhoeffer, executed by the Nazis on 9 April 1945, just a month before the end of World War II in Europe. In June 1944, Bonhoeffer wrote a poem which showed he was plainly wrestling with questions of identity. His plaintive question *Who Am I?* also testifies to the necessity of a faith—and identity—grounded ultimately in God alone:

> Who am I? They mock me these lonely questions of mine.

Whoever I am, you know me, O God. You know I am yours.[31]

When all the identity supports had been torn from his life, as he awaited in prison his fate at the hands of the Nazis, Bonhoeffer's identity was grounded not in himself. No doubt his faith was nurtured in a lifelong commitment to God, with strong intellectual, practical, and affectional components. Foundational, however, is that he is known by God and that he belongs to God. Assured of God's love, he could rest in divine grace free from the need to establish his own identity. We, too, are known by God and belong to him for eternity. This is not merely our hope, it is also our identity, for we belong amongst the community of those who have been baptised.

**Funding:** This research received no external funding.

**Conflicts of Interest:** The author declares no conflicts of interest.

## Notes

1  See "The Encyclical Epistle of the Church at Smyrna Concerning the Martyrdom of the Holy Polycarp" in (Roberts and Donaldson 1885, p. 41).

2  A recent volume of forty two essays indicates the variety of approaches to Christian identity formation; see (Houston and Zimmermann 2018).

3  (Crawford and Rossiter 2006, p. 90). Compare the eight factors identified by Snodgrass (2011a, pp. 11–14).

4  (Crawford and Rossiter 2006, p. 23). Crawford and Rossiter cite (Meijer 2006, pp. 92–99).

5  (Esler 2014, p. 19). Michael Hogg defines SIT: "Social identity theory is a social psychological analysis of the role of self-conception in group membership, group processes, and intergroup relations". See (Hogg 2018).

6  (Roccas and Brewer 2002, p. 91). The authors' description of the merger model is not straight-forward. In the diagrammatic representation of the models provided in the article, two partly overlapping identities are portrayed as roughly equivalent *against the background of* or *enclosed within* a larger amorphous and unidentified field. Later, they suggest that the model requires "the introduction of some superordinate principle that makes the inconsistent cognitions compatible" (also page 91). Given the nature of the case, the superordinate principle also remains unnamed, but one wonders whether it might be akin to Kwame Anthony Appiah's idea of the 'cosmopolitan'—which we note below.

7  (Esler 2014, pp. 32, 35). So also (Hogg 2018): "People construct group norms from appropriate in-group members and in-group behaviors and internalize and enact these norms as part of their social identity". As such, conformity is not "surface behavioral compliance but a deeper process whereby people's behavior is transformed to correspond to the appropriate self-defining group prototype".

8  (Appiah 2018, p. 219). Appiah concludes his book by commending the famous quote of Terence (Publius Terentius Afer): "I am human, I think nothing human alien to me" (p. 219).

9  It is beyond the scope of this essay to provide a theological account of baptism itself, and the various critical questions pertaining to the topic. For studies, see (Beasley-Murray 1973; Schreiner and Wright 2007; Ferguson 2009; Streett 2018).

10  For a discussion of these ideas see (Davids 1990, pp. 11–14; Jobes 2005, pp. 53–56). See also Miroslav Volf's essay "Soft Difference: Church & Culture in 1 Peter" in (Volf 2010, pp. 65–90).

11  Thesis 28 of Luther's Heidelberg Disputation is relevant here: "God's love does not find, but creates, that which is pleasing to it". See (Wengert 2015, pp. 85, 104–5).

12  See (Barth 1956). For a discussion of Barth's lecture, see (O'Neil 2013, pp. 75–82).

13  An adaption of the first article of the Westminster Catechism. To claim this as the chief aim of human existence is not to deny the possibility of a vision of human flourishing. In a recent book, Miroslav Volf makes much of Jesus' statement in John 6:51 that Jesus came as the bread from heaven to give his life "for the life of the world". Volf argues from this for a vision of 'flourishing life'. Evident in Volf's exposition is a concern for the flourishing of all of life and not merely the flourishing of one's own life. See (Volf and Croasmun 2019).

14  Compare 2 Corinthians 5:16–17 where Paul can say, "henceforth we recognise no one according to the flesh...if anyone is in Christ they are a new creature". See also his setting aside of identity markers in Philippians 3:3–8.

15  For a thoughtful consideration of this issue, see (Ezigbo 2018).

16  See also (Du Toit 2023). In his essay, Du Toit correctly notes that Christian identity is divinely grounded rather than socially constructed, and that Christians are 'foreigners' with respect to social environments. This does not annul the usefulness of SIT as a descriptive and heuristic tool in biblical studies but does indicate something of its limits: it cannot provide a normative account of Christian identity formation.

17  This remains the case even without implying doctrines such as original sin or total depravity.

[18]　(Barth 1969, pp. 200–1). Mark Lindsay notes that for Barth participation in this mission cannot be reduced to proclamation alone. Rather, "mission also and necessarily includes within it those acts of ethical and political solidarity with others, by which the entirety of human life is made more *human*". See (Lindsay 2013, p. 243, original emphasis). See also James Smith's contention that baptism functions as ordination did in the Old Testament with believers being brought into the priesthood and ministry of Christ: (Smith 2009, p. 184).

[19]　Again, as noted above, I cannot in this essay discuss in detail the theology underpinning this assertion. It is evident to anyone familiar with the topic that every Christian tradition and theologian brings their own nuance to the matter, some laying more emphasis on (or even denying!) either the former or the latter aspect of the assertion. See, as an example, (Hunsinger 2000).

[20]　(Fee 1994, pp. 860–64). James Dunn arrives at a similar conclusion although he allows a little more space for the development of a sacramental understanding of baptism in his helpful exposition. See (Dunn 1998, pp. 413–59; especially 419, 425, 452–53, 456).

[21]　See, for example, (Snodgrass 2011b, p. 167). See also Jesus' baptism in Matthew 3:13–17 (cf. Luke 3:21–22), and those of Paul, Cornelius, and the Ephesians in Acts 9, 10, and 19 as well as Acts 2:38–39, 1 Cor. 6:11; and Titus 3:4–6. Some commentators view Jesus' teaching about being born from above (John 3:1–10) as referring to baptism. See, for example, (Ridderbos 1997, pp. 127–28; Bruner 2012, pp. 175–78, 181–88).

[22]　(Dunn 1998, pp. 442–47). The methodological point is worth noting: if the 'traditional' reading of Paul tended to 'over-read' baptismal contexts and theology into Paul's expressions, Dunn tends towards a caution that will seek an alternative to baptismal interpretation unless it is explicit in the text.

[23]　This is my own adaption of lists presented by Fee, Dunn, and Lane. See (Fee 1991, p. 117; Dunn 1998, p. 456; Lane 2020, pp. 93–111). Fee claims that early Christian experience of the Spirit in conversion was both dynamic and usually visible; see (Fee 1994, pp. 863–64).

[24]　(O'Loughlin 2011, p. 84). Note that O'Loughlin's claim is concerned with the establishment of norms, or more precisely, the processes by which norms are identified and adopted and the inputs feeding these processes, than with navigating or negotiating identity complexity per se. The point is to 'become a Christian' and the result is that modes of Christian life and identity reign supreme in the life of the novice. See also (Kreider 2016, p. 139).

[25]　For an excellent discussion of reading Scripture as an ecclesial practice, see (Verhey 2002, pp. 49–76).

[26]　The ESV translates *kai hē euprepeia tou prosōpou autou apōleto* (Jas. 1:11) as 'and its beauty perishes'. Literally the phrase is 'and the beauty of its face perishes'. The NASB translates as 'the beauty of its appearance is destroyed', retaining the genitive, but losing the personification of the image ('its beautiful face is destroyed'), and its resulting power when applied to the rich person. Note that in Jas. 2:1 the new community are instructed not to 'receive the face' (*prosōpolēmpsiais*) of the rich man but to accord equal honour to rich and poor alike, and indeed, to privilege the poor as God has done (2:5).

[27]　(O'Loughlin 2011, pp. 79, 81). The imagery is also prominent in the work of Stanley Hauerwas. See (Hauerwas 1991; 1999, pp. 93–111).

[28]　See Eugene Peterson's book bearing this title. Peterson credits the phrase to Friedrich Nietzsche: (Peterson 1980, p. 13).

[29]　(O'Loughlin 2011, p. 81); cf. Didache 7:1, on page 88.

[30]　See (Butterfield 2014, pp. 19–20, 38–41). Also note Carswell's observation that increasing secularisation in the West means that even churches of paedobaptist traditions will find that the numbers of infant baptisms in their congregations has declined sharply, as has the proportion of babies in the community being baptised. Such churches are again in conditions in which their mission will require a greater emphasis on evangelism and thus believer's baptism. Carswell insists that a process such as the RCIA is an integral aspect of such evangelisation. (Carswell 2018, pp. 431–32).

[31]　The final couplet of Bonhoeffer's "Who Am I?" (Kelly and Nelson 1995, p. 514).

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
