# Peer review of "The Role of Baptism in Christian Identity Formation"

_religions, doi:10.3390/rel15040458_

Round 1

Reviewer 1 Report

Comments and Suggestions for Authors

The article has a good structure and addresses a topic of interest in the church.

It would be strengthened by some more discussion on social identity theory. It would be good to see some references to the T& T Clark Handbook on Social Identity as this is a valuable work in the field of social identity.

The author should add something to this about the complexity of identity, which would be particularly relevant for how Christian identity gives a new identity. See the work of Roccas and Brewer on the subject or the work of Kobus Kok.

Consider adding this article by Roccas and Brewer: https://doi.org/10.1207/S15327957PSPR0602 or this one by Kobus Kok DOI: 10.4102/hts.v70i1.2708.

or this: Du Toit, P.L.G. A Foreign People: Towards a Holistic Identity Theory within a Christian Context. Religions 202314, 1167. https://doi.org/10.3390/rel14091167

Eliminate expressions like "willy nilly." This is a colloquialism.

Author Response

Thank you for your helpful comments. I have now incorporated a section on Social Identity Theory in the first section of the essay (the survey of contemporary philosophy on identity), including those sources recommended, and then used this material as appropriate in the reflections in the remainder of the essay. I removed 'willy nilly' and cleaned up several other errors in the manuscript.

Reviewer 2 Report

Comments and Suggestions for Authors

1.      Because the author deals with social identity, it is imperative that he cite Henri Tajfel and John Turner, pioneers in the field. The author does NOT reference them.

2.      Because the author deals with identify from a biblical perspective, he should interact with Philip Esler and Brian Tucker, both of whom have written extensively on the topic. Unfortunately, the author does NOT enter into a conversation with them.

3.      Because the author deals in particularly on the topic of baptism and identity s/he needs to reference Everett Ferguson and R. A. Streett, whose scholarly treatments are essential. No references can be found in the article.

4.      Redundancy—Author twice uses the phrase “gift of grace” (pp. 1, 13). A gift by its nature is offered by grace.

5.      Author uses the phrase “martyrs’ fate.” It should be “martyr’s fate.” (p. 1, para. 2)

6.      Spelling mistake. Author has “decentring” instead of “decentering.” (p. 9)

7.      The author writes from a subjective viewpoint as a “Christian” and “Baptist, rather than as objective scholar (to which s/he admits, p. 19). This colors the entire essay. As a result, the author tends to theologize.

8.      The author promotes baptism in the name of the triune God as the vehicle of Christian social identity, despite the trinitarian formula being mentioned only once in the Christian Bible (Matt. 28). All other mentions of baptism, which s/he does not reference, speak of baptism in Jesus’s name.

The author is prescriptive rather than descriptive in his/her treatment of the texts, making the essay more pastoral and sermonic and aimed at a Christian audience rather than scholarly and aimed at academics.

Author Response

Thank you for your helpful comments. I have added a section on Social Identity Theory to the first section of the essay, including discussion of those sources recommended, and then used this material as a basis for reflection in the latter two sections as appropriate. The already-long essay is almost 2000 words longer as a result. But I couldn't simply reference SIT without discussion and reflection.

With respect to your third point, I did explicitly note that the scope of this essay did not allow such discussion. However, I have added the suggested references to the section on baptism for those interested in studying that topic further. 

I have cleaned up the various typos indicated, among others, although martyrs' remains as it was, since the reference is plural in that sentence, as in the following sentence.

Item 8 on the list falls under the footnote mentioned above about a discussion of the doctrine of baptism, falling outside the scope of this essay. It is true that the NT speaks of baptism 'in Jesus' name' or 'into Christ' or some similar expression more often than it does baptism in the name of the triune God. But I did not feel that this discussion was ever the focus of this paper, important though the point is. I trust that the footnote concerning the doctrine and associated critical questions will be sufficient. 

It is true that I write as a Baptist. I included the brief concluding comments precisely because I am aware that my discussion reflects my own identity and location - it seemed relevant in an essay on this topic! I was also aware that my position is not the only position on the topic in the Christian church. For this reason I wished to comment briefly on the relevance of the study for those in traditions who do affirm and practise infant baptism. I did consider moving these comments to the introduction but chose against it because (i) it seemed to break the flow from the introduction into the substance of the essay, and (ii) the comment about the relevance of the study to infant baptism seemed better placed after the preceding discussion. This could be a purely subjective judgement, but it was made after consideration of the alternative. 

It is difficult to know precisely how to respond to the second sentence of point 7 and the final section of point 8, both of which are related. I must confess that yes, I am an ecclesial theologian, having spent more than twenty years of my adult life in the pastorate. I have a natural orientation in my research and writing toward the end of serving the church. I also have concerns about the notion of 'scholarly objectivity': on the one hand I concur that research and scholarship must deal truthfully with the sources and evidence it engages, and work within careful and explicit methodological frameworks; I am not sure that theology can be 'objective' in the sense sometimes used (I do not impute this to the reviewer - our discussion is too brief for the kind of nuance required in a proper conversation), that is, as though one can step outside of oneself, and indeed, outside of the bounds of faith. I have been shaped in these convictions with a Barthian epistemology, and I know it shows. I do believe, however, that the position is defensible.

Again, thank you for your suggestions: I feel that inclusion of reflection on Social Identity Theory, although brief, has strengthened the essay, and perhaps it may also allay some of your concerns with the material as presented.

Warm regards,

Michael.

Round 2

Reviewer 1 Report

Comments and Suggestions for Authors

Paper is acceptable now that it has been revised.

Reviewer 2 Report

Comments and Suggestions for Authors

The revised article is much improved and ready for publication.